# Systemic Antibiotic Therapy in Hidradenitis Suppurativa: A Review on Treatment Landscape and Current Issues

**DOI:** 10.3390/antibiotics12060978

**Published:** 2023-05-29

**Authors:** Elisa Molinelli, Edoardo De Simoni, Matteo Candelora, Claudia Sapigni, Valerio Brisigotti, Giulio Rizzetto, Annamaria Offidani, Oriana Simonetti

**Affiliations:** Dermatological Unit, Department of Clinical and Molecular Sciences, Polytechnic Marche University, 60126 Ancona, Italygrizzetto92@hotmail.com (G.R.); annamaria.offidani@ospedaliriuniti.marche.it (A.O.); o.simonetti@staff.univpm.it (O.S.)

**Keywords:** antibiotics, antibiotic resistance, clindamycin, dalbavancin, ertapenem, guidelines, hidradenitis suppurativa rifampicin, tetracycline

## Abstract

Hidradenitis suppurativa (HS) is a chronic, recurrent, and inflammatory skin disease characterized by painful, deep-seated, nodules, abscesses, and sinus tracts in sensitive areas of the body, including axillary, inguinal, and anogenital regions. Antibiotics represent the first-line pharmacological treatment of HS because of their anti-inflammatory properties and antimicrobial effects. This narrative review summarizes the most significant current issues on the role of systemic antibiotics in the management of HS, critically analyzing the main limits of their use (antibiotic resistance and toxicity). Although, in the last decades, several cytokines have been implicated in the pathomechanism of HS and the research on the use of novel biologic agents in HS has been intensified, antibiotics remain a valid therapeutic approach. Future challenges regarding antibiotic therapy in HS comprise their use in association with biologics in the management of acute flare or as a bridge therapy to surgery.

## 1. Introduction

Hidradenitis suppurativa (HS) is a chronic inflammatory skin disease of the pilo-sebaceous unit, affecting intertriginous areas, which negatively affects the quality of life. The prevalence ranges from 1% to 4%, with an estimated ratio of males and females of 1:3 in North American and European patients [1,2]. Characteristic disease lesions include recurrent and painful nodules, abscesses, draining fistulas, and irreversible fibrotic scars [3].

Because of the lower specificity of initial clinical signs and the low awareness in physicians, HS is burdened by a diagnostic delay of about 7 years [4,5]. Early recognition and adequate treatments are critical to improving the prognosis and the quality of life of patients with HS [6,7].

The diagnosis of HS is based on the clinical finding of typical HS lesions located predominantly in the intertriginous areas (most commonly the axillae, the groin, and anogenital regions) and can be complemented by radiological imaging (high-frequency ultrasonography and magnetic resonance imaging) and histopathological features [8,9]. The assessment of HS severity and disease burden can be measured using several tools including hidradenitis suppurativa physician’s global assessment (HS-PGA) scale, Hurley scoring system, international hidradenitis suppurativa severity score system (IHS4), pain visual analogue scale (pain VAS), numeric rating scale (NRS) for pain, hidradenitis suppurativa clinical response (HiSCR), and dermatology life quality index (DLQI) [10,11,12] (Table 1).

The exact pathophysiology of HS is not completely understood. However, follicular hyperkeratosis, occlusion, rupture, and secondary bacterial colonization with biofilm formation have a key role in the development of the disease [3,13,14]. In several solid tumors and lymphomas, it was demonstrated that an infectious agent may initiate chronic inflammation, and therefore, infection could be the background of lymphocyte transformation and lymphomagenesis. Although HS is not primarily caused by bacterial infection, it is possible to speculate that the polymicrobial microflora colonizing HS lesions would seem to exert a key role in the vicious cycle of inflammation [15,16]. The principal bacterial species found in HS lesions included coagulase-negative staphylococci, mixed anaerobic bacteria, *S. aureus*, and streptococcal species [13,17,18].

Antibiotics represent the first-line treatment of HS, mainly because of their anti-inflammatory properties in association with their antimicrobial and immunomodulatory effects. The activity and efficacy were demonstrated in several observational and interventional studies. However, this treatment approach is burdened by the occurrence of acquired resistance [6,19,20].

This narrative review aims to summarize the use of systemic antibiotic treatment in HS, highlighting the most significant current issues and future challenges on their use in the management of the disease.

## 2. Materials and Methods

Bibliographic searches for qualitative review were conducted in PubMed up to 20 February 2023, with no date limits, by using the terms: (hidradenitis suppurativa OR acne inversa OR Verneuil’s disease) AND (antibiotic), (lincosamides), (clindamycin), (rifampicin), (tetracycline), (lymecycline), (doxycycline), (beta-lactam antibiotics), (penicillins), (cephalosporins), (carbapenems), (monobactams), (ertapenem), (macrolides), (azithromycin), (clarithromycin), (erythromycin), (metronidazole), (glycopeptide), (dalbavancin), (linezolid). One author (EDS) initially screened all titles and abstracts and excluded articles that were clearly ineligible. Reports and cases were excluded if clinical details were lacking. When eligibility was in doubt, the other two authors (EM) and (MC) were involved. Articles were limited to those in the English language. Full texts of the included articles were reviewed, and reference lists were manually searched and were checked for additional sources.

## 3. Current Antibiotic Therapy of HS

Systemic antibiotics have been a mainstay of HS treatment for decades, with different reported regimens. Monotherapy is normally used for mild-to-moderate disease, whereas in severe and advanced disease their role is adjunctive on account of lower response rates and increased recurrence. The antibiotics used in HS are discussed according to their mechanisms of action (Table 2; Figure 1).

### 3.1. Lincosamidess

Lincosamides are bacteriostatic antibiotics that inhibit the 50S ribosomal subunit and alter the structure of the bacterial biofilm by modifying polysaccharide synthesis [21]. They include clindamycin, which is active against Gram-positive cocci (except *Enterococci*) and anaerobes [22]. Moreover, lincosamides showed anti-inflammatory properties, modulating the expression of nuclear factor kappa-light-chain enhancer of activated B cells (NF-kB) and activator protein 1 (AP-1) genes, and decreasing the activity of tumor necrosis factor-alpha (TNF-α), interleukin-8 (IL-8), interleukin-6 (IL-6), and granulocyte macrophage colony-stimulating factor (GM-CSF) in neutrophils [4].

Clindamycin is indicated in HS, in topical 1% or systemic formulation, alone or in combination with other antibiotic agents. Topical clindamycin 1% applied twice daily for 12 weeks represents the first-line therapy in mild-to-moderate HS. Although topical clindamycin is frequently prescribed, data supporting its efficacy are limited. In a double-blind trial, 27 patients with axillary and/or inguinal HS were randomly assigned to receive topical clindamycin 1% or topical placebo for three months. Analysis of cumulative disease burden score showed greater improvement in the clindamycin group than in the placebo group, after one, two and three months of treatment [23].

Moreover, topical clindamycin was compared with systemic tetracyclines in a double-blind clinical trial enrolling 46 patients with stage I–II HS; patients were randomized to receive systemic tetracycline 1 g/day or topical clindamycin 1% for 3 months and no significant differences were found in terms of regression of abscesses and nodules [24].

Topical clindamycin 1% was compared in patients with mild-to-moderate HS to topical resorcinol 15%, a topical chemical peeling agent with keratolytic and anti-inflammatory properties. In a retrospective study of 134 patients with mild-to-moderate HS, resorcinol was demonstrated to be an excellent alternative to topical clindamycin in the acute flare and maintenance therapy of HS, reducing lesions in both number and size and significantly prolonging disease-free survival [25,26].

In addition, in an open study of 12 patients with stage I or II HS evaluating the activity of topical resorcinol 15% once to twice daily mainly during disease flare, all patients experienced a reduction in pain and duration of painful abscesses [27].

The combination of clindamycin (300 mg bis in day [BID]) and rifampicin (300 mg BID) for 10–12 weeks may be considered in patients not responding to oral tetracyclines or, as first-line treatment, for patients with a significant inflammatory burden (moderate-to-severe HS). Rifampicin is a bactericidal antibiotic that showed activity against *Mycobacteria*, Gram-positive and -negative bacteria, by inhibiting DNA-dependent RNA-polymerase, and demonstrated a reduction in NF-κB activity and Th17 differentiation, and modulated T-cell responses [28]. In addition, due to its induction effect on cytochrome 450 (CYP450) 3A4, rifampicin interacts with the metabolism of several drugs, and determines a decrease in the bioavailability of the clindamycin. However, monotherapy with rifampicin is usually contraindicated considering the risk of emergence of antibiotic resistance in *Mycobacteria* strains in patients with latent tuberculosis [29].

A retrospective study evaluating the activity of the combination of clindamycin and rifampicin for 10 weeks in 116 patients with severe HS, reported a marked improvement of Sartorius score from the enrollment to the end of treatment (median Sartorius score 29 vs. 14.5; *p* < 0.001) [28].

In a retrospective study of 34 patients with HS, the maximum treatment effect was shown to occur within 10 weeks of therapy; following total remission, 8 of 13 (61.5%) patients experienced a relapse after a mean period of 5.0 months [30].

In addition, a prospective study of 26 patients with HS evaluated the efficacy, safety, and relapse rate of HS treated with the combination of clindamycin 600 mg and rifampicin 600 mg daily for 12 weeks. Patients were followed for 1 year. After 12 weeks, a clinical response was observed in 19 patients (73%); the response was positively related to the female sex (*p* = 0.02) and not with body mass index, Hurley stage, or lesion location. At a 1-year follow-up, efficacy was maintained in 7 (41%) patients, while 10 (59%) relapsed after a mean time of 4.2 months. The data suggest that oral clindamycin with oral rifampicin for 12 weeks is an effective and tolerable regimen for HS [31,32,33].

The benefit derived from the addition of rifampicin was not prospectively compared with clindamycin alone. Only two retrospective and therefore *a priori* biased comparative studies are available: they included 60 and 53 patients with mild–moderate–severe HS and both excluded significant differences in terms of efficacy between the combination and clindamycin monotherapy. Moreover, greater efficacy of clindamycin alone on fistulas was demonstrated. It has been hypothesized that the lower concentration of clindamycin resulting from the inhibitory effect of rifampicin may reduce the efficacy of the treatment in severe HS lesions such as fistulas, which are often colonized by a polymorphous and abundant anaerobic flora [32].

Clindamycin was also evaluated in association with oxofloxacin, a second-generation fluoroquinolone which acts to inhibit topoisomerase II in Gram-negative bacteria and topoisomerase IV in Gram-positive bacteria. Specifically, it is active against *Staphylococcus* spp. and anaerobic bacteria. The rationale for substituting oxofloxacin with rifampicin is to reduce rifampicin resistance and reserve it for cases of Methicillin-resistant Staphylococcus aureus (MRSA) and Methicillin-sensitive Staphylococcus aureus (MSSA) infection. A recent retrospective study including 65 patients with HS showed that the combination of clindamycin (600–1800 mg according to weight) and oxofloxacin (200–400 mg) was associated with a clinical improvement in 58% of patients, with a complete response observed in 33.8% of patients and partial remission in 24.6% of patients [4,34].

### 3.2. Tetracyclines

Tetracyclines exert their antibiotic activity by binding the 30S subunit of bacterial ribosomes, reducing proinflammatory cytokines (e.g., IL-1, IL-6, and IL-8), which are increased in patients with HS, and promoting the secretion of anti-inflammatory cytokines such as IL-10 [4].

Tetracyclines are characterized by a broad spectrum of action toward both aerobic and anaerobic Gram-positive and Gram-negative bacteria. They include tetracycline, doxycycline, lymecycline; according to their activity HISCR, they are recommended for Hurley stage I or early-stage II HS [35].

A prospective study including 108 patients with HS demonstrated the efficacy and safety of tetracyclines (tetracycline 500 mg BID, doxycycline 100 mg BID, and lymecycline 300 mg BID) in the treatment of HS. The mean hidradenitis suppurativa score (HSS) at baseline was 26.10 (SD 20.18) points, improving to 17.97 (SD 17.88) at follow-up, difference is 8.13 (95% CI 5.21–10.93), *p* < 0.0001. The highest improvement in HSS was observed in the tetracycline group. All patients showed a statistically significant improvement of HR severity, and the most significant clinical improvement was observed in the tetracycline group [36].

A prospective study including 20 patients investigated the efficacy of the combination of minocycline (100 mg day) plus colchicine (0.5 mg BID) for 6 months, followed by maintenance with colchicine for 3 months. Colchicine is a toxic natural product, frequently used in the treatment of gout that acts by inhibiting the polymerization of microtubules, and then numerous inflammatory pathways. All patients showed improvement within the first 3 months of therapy and continued to improve over the next 6 months. None presented a worsening of pathology during this period or discontinued the treatment for adverse events [37].

In a retrospective study with 52 patients, lymecycline monotherapy was also compared with the clindamycin–rifampicin combination in terms of clinical response at the end of the antibiotic treatment period, according to the HiSCR. A significant improvement of disease activity was observed in both groups of patients examined. In both studies, the greatest results were obtained with nodules and abscesses rather than fistulas, regardless of the site involved. Nodular-type HS seems to respond better to lymecycline, whereas the abscess/tunnel type seems to respond better to clindamycin plus rifampicin [20].

### 3.3. Beta Lactam Antibiotics

Beta-lactam antibiotics are bactericidal agents interfering with bacterial cell wall synthesis. They are classified according to resistance development and spectrum of action, namely penicillins, cephalosporins, carbapenems, and monobactams [38].

Ertapenem is a broad-spectrum carbapenem [39]. A retrospective study based on 30 patients with severe HS receiving ertapenem (1 g/day) demonstrated a 50% reduction in Sartorius score, a rapid improvement in quality of life, and clinical remission of 67% of Hurley I nodules and 26% of Hurley II lesions [40].

In a second retrospective study including 36 patients with Hurley II and III HS, ertapenem administration was associated with a clinical improvement in 97.2% of patients and an improvement in quality of life in 85.7% of patients [3].

Ertapenem has a potent effect against anaerobic bacteria whose resistance in community infections is low (<1%) and has been reported only in nosocomial infections. Therefore, it can be used as an empirical treatment in polymicrobial infections, in HS refractory to other therapies, or when surgery is contraindicated. From a safety point of view, ertapenem appears to be safe [4,10]. Carbapenems may cause an elevation of transaminases, but this decreases with the discontinuation of treatment. Rarely, the elevation of cholestasis enzymes has been reported, especially in patients with numerous comorbidities or other liver diseases [41].

### 3.4. Macrolides

Macrolides, which include azithromycin, clarithromycin, and erythromycin, are inhibitors of protein synthesis and exert their bactericidal activity by inhibiting the ribosomal 50S. Macrolides are characterized by anti-inflammatory and immunomodulatory effects, interacting with phospholipids and transcription factors AP-1, NF-kB, and other inflammatory cytokines [42].

In a retrospective study including eight prepubertal patients with HS, oral azithromycin (10 mg/kg/day for 3 consecutive days) in combination with topical clindamycin was evaluated for the treatment of acute flares. This association showed a significant reduction in pain VAS, IHS4, and DLQI compared to patients treated with topical clindamycin and oral zinc supplementation (90 mg/day) [11,12,43,44].

However, the treatment of HS with macrolides, and in general with traditional antibiotics, is limited by the presence of increasing data on bacterial resistance to macrolides, lincosamides, and streptogramin B (MLSB) [45,46]. In addition, beneficial results in the management of adult patients with moderate-to-severe HS were obtained by the combination of macrolides and acitretin [47].

### 3.5. Metronidazole

Metronidazole is a nitroimidazole compound, used in patients with HS as part of a triple broad-spectrum antibiotic therapy, in combination with rifampicin and moxifloxacin, for 4–6 weeks [48].

In two studies, the administration of this triplet was associated with a rapid and complete response. However, it is difficult to establish the contribution of metronidazole in this association strategy. It has been hypothesized that Hurley I or II HS flare-ups might have a good and rapid response to metronidazole (550 mg ter in day TID] for 2 weeks). The short antibiotic course could reduce the overall resistance in HS and flare-up by eradicating the anaerobic bacterial load. In addition, metronidazole is active against *Provotella* spp., which is increasingly resistant to clindamycin [49,50].

Another hypothetical partner of metronidazole is ceftriaxone, a 3rd-generation cephalosporin: the combination of metronidazole and ceftriaxone, as an induction regimen, demonstrated to provide a clinical remission in four patients with HS [4,51].

**Table 2 antibiotics-12-00978-t002:** Studies evaluating the activity of systemic antibiotics in HS.

Treatment	Authors	Study	N	Patients’ Characteristics	Study Design Endpoints	Outcomes
**CLINDAMYCIN (Monotherapy and combinations)**						
Clindamycin [23]	Clemmensen O.J. et al., 1983	Double-blind trial	30	Hurley stage not specified	Clindamycin vs. placebo	Clindamycin significantly superior to placebo except for inflammatory nodules and abscesses at each monthly evaluation (*p* less than 0.01)
Clindamycin [24]	Jemec G.B.E. et al., 1998	Double-blind placebo-controlled RCT	46	Hurley stage I or II HS	Tetraciclyne 1 g/die PO + topical placebo vs. placebo PO + topical clindamycin 1% for at least 3 mo	No significant differences between the two treatments in terms of VAS score Patients’ global assessment of disease was significantly worse than physician’s assessment in 3 of 5 evaluations (*p* = 0.0096 to 0.015), but the correlation between patients’ and physicians’ assessments was satisfactory after only one visit (rs = 0.761 to 0.895)
Clindamycin [25]	Molinelli E. et al., 2020	Retrospective study	124	Hurley stage I or II HS	Topical clindamycin 1% (N = 739). Topical resorcinol 15% (N = 61).	Patients treated with resorcinol 15% showed a significant improvement in HS clinical response, international HS severity core system and pain visual analogue scale score from baseline compared to patients treated with clindamycin; In group A (clindamycin 1%), clinical response (HiSCR) was obtained in 38 (52%) of 73 patients after 12 weeks (*p* < 0.01). In group B (resorcinol 15%), clinical response was achieved in 52 (85.3%) of 61 patients after 12 weeks (*p* < 0.001). At 12 weeks, the clinical response to resorcinol 15% was higher than the response to topical antibiotic, with statistically significance (*p* < 0.001).
Clindamycin [32]	Caposiena Caro R.D. et al., 2019	Retrospective study	60	Hurley stage I or II HS	Group A: clindamycin 150 mg 4 times a day + rifampicin 300 mg BID. Group B: clindamycin 150 mg 4 times a day.	After 8 weeks of treatment the responses to antibiotics were similar in both groups.
Clindamycin [13]	An J.H. et al., 2021	Retrospective study	53	Hurley stage II or III HS	Clindamycin monotherapy for 8 weeks	Improvement in rate of HS clinical response (Hi-SCR) achievers and comparing HS physician’s global assessment (HS-PGA) before (W0) and after (W8) the treatment. Twenty-one patients (61.76%) achieved Hi-SCR. The mean scoring of HS-PGA had significantly decreased from 3.24 to 2.15 (*p* = 0.001).
Clindamycin-rifampicin [28]	Gener G. et al., 2009	Retrospective study	116	Hurley stage III HS	Clindamycin (300 mg PO BID) + rifampicin (600 mg PO daily)	Improvement of median Sartorius score at the end of treatment (29 vs. 14.5; *p* < 0.001), and of QoL score.
Clindamycin-rifampicin [30]	van der Zee H.H. et al., 2009	Retrospective study	34	Hurley stage I, II, III HS	Clindamycin (300 mg PO BID) + rifampicin (600 mg PO daily)	Partial improvement: 82%; total remission: 47% (maximum effect of treatment within 10 weeks). Following total remission, 8 of 13 (61.5%) patients treated as mentioned above experienced a relapse after a mean period of 5.0 months.
Clindamycin-rifampicin [31]	Dessinioti C. et al., 2016	Prospective study	26	Hurley stage I, II, III HS	Clindamycin (300 mg PO BID) + rifampicin (600 mg PO daily) for 12 weeks	12-week clinical response rate: 73%; At the 1-year follow-up, there was sustained efficacy in 7 (41%) patients, while 10 (59%) had disease relapse after a mean time of 4.2 months.
Clidamycin-Oxofloxacin [34]	Delaunay J. et al., 2018	Retrospective study	65	Hurley stage I, II, III HS	Clindamycin (600–1800 mg) + oflaxacin (200–400 mg)	Efficacy in disease control in HS. Thirty-eight patients (58.4%) reported improvement of disease activity under OC with complete response for 22/65 (33.8%) and partial remission in 16/65 (24.6%) patients.
**TETRACYCLINES**						
Minocycline [37]	[37] K. et al., 2017	Prospective study	20	Hurley stage I, II, III HS	Minocycline 100 mg PO + colchicine 0.5 mg BID for 6 months, followed by colchicine maintenance 0.5 mg BID for 3 months	Efficacy in disease control in HSPhysician global assessment (PGA) scale (PGA) shows good and excellent response in 95% of patients at 9 months.
Limecycline [20]	Caposiena Caro R.D. et al., 2021	Retrospective study	52	Hurley stage I, II, III HS	Group A (N = 26): lymecycline 300 mg daily for 10 weeks. Group B (N = 26): clindamycin 600 mg + rifampicin 600 mg for 10 weeks.	Both treatments are effective in terms of IHS4, pain VAS scale and DLQI for patients with moderate-severe HS. Response rates at the end of the treatments were similar in both groups (*p* = 0.78): 57.7% in group A and 53.8% in group B met the primary outcome (HiSCR).
Tetracycline [36]	Jørgensen A.R. et al., 2021	Prospective study	143	Hurley stage I, II, III HS	Tetracycline 500 mg PO BID. Doxycycline 100 mg PO BID. Lymecycline 300 mg PO BID.	All treatments were effective and safe in HS patients. Tetracycline provided the greatest clinical improvement mean The mean HSS at baseline was 26.10 (SD 20.18) points, improving to 17.97 (SD 17.88) at follow-up, difference is 8.13 (95% CI 5.21–10.93), *p* < 0.0001 sured by HSS.
**ERTAPENEM**						
Ertapenem [40]	Join-Lambert O. et al., 2016	Retrospective study	30	Hurley stage III	Ertapenem 1 g die for 6 weeks + antibiotic consolidation treatments for 6 months (M6) in severe HS	Dramatic improvement of severe HS provided by ertapenem. The median (IQR) Sartorious score dropped from 49.5 (28–62) at baseline to 19.0 (12–28) after ertapenm (*p* < 10^−4^).
Ertapenem [3]	Braunberger T.L. et al., 2018	Retrospective study	36	Hurley stage II, III HS	Ertapenem 1 g die	Clinical improvement in 97.2% of patients and an improvement in quality of life in 85.7% of patients.
**METRONIDAZOLE (monotherapy and combinations)**						
Metronidazole [50]	Delage M. et al., 2023	Prospective trial	28	Hurley stage I HS	Rifampicin + moxifloxacin + metronidazole	The primary endpoint was a Sartorius score of 0 (clinical remission) at week 12. The median Sartorius score dropped from 14 to 0 (*p* = 6 × 10^−6^) at week 12, with 75% of patients reaching clinical remission.
Ceftriaxone + Metronidazole [51]	Nassif A et al., 2012	Case report	4	Hurley stage II HS	Ceftriaxone IV + metronidazole PO	Improvement of HS
**DALBAVANCIN**						
Dalbavancin [6]	Molinelli E. et al., 2022	Retrospective study	8	Hurley stage I, II or III HS	Dalbavancin 100 mg IV	Significant disease improvement at 12 weeks; Significant disease improvement was achieved at 12 weeks (T12) with average values of 7 for IHS4, 2 for pain VAS, and 8 for DLQI, and HiSCR was satisfied in six out of eight patients compared to baseline (T0)

Keys: BID: bis in die; Hi-SCR: Hidradenitis suppurativa clinical response; HS: hidradenitis suppurative; mo: months; N: number; HS-PGA: Hidradenitis suppurativa-physician’s global assessment; IHS4: Hidradenitis suppurativa severity score system; PO: per os; RCT: Randomize clinical trials; VAS: Visual analogue scale.

### 3.6. Dalbavancin

Dalbavancin is a semisynthetic glycopeptide exclusively used for the treatment of acute skin infections caused by Gram-positive, including methicillin-resistant and methicillin-sensitive *S. aureus*, *S. pyogenes*, *S. agalactiae*, and *E. faecalis* [52]. These microorganisms are closely associated with the HS microbiota. Dalbavancin in addition to its antibacterial effect has also been shown to stimulate faster tissue improvement and repair processes by reducing the expression of Matrix metallopeptidase-1 (MMP-1) and Matrix metallopeptidase-9 (MMP-9) and increasing the expression of Epidermal Growth Factor Receptor (EGFR) and Vascular-Endothelial Growth Factor (VEGF) [53,54].

In a study of eight patients with moderate-to-severe HS with isolated *Staphylococcus aureus*, *Streptococcus agalactie*, *Enterococcus fecalis*, and *Proteus* collected from purulent material on HS lesions, efficacy, flare, and disease-free survival after dalbavancin therapy were evaluated. Significant disease improvement was obtained at 12 weeks (T12) with average values of 7 for IHS4, 2 for pain VAS and 8 for DLQI achieved in six of the eight patients compared to baseline (T0) [6,11,12].

Currently, dalbavancin has a role in selected cases: in the control of disease acute flare; as a bridge therapy to surgery; in the support of biologic therapy with adalimumab or other off-label biologic therapy; in case of loss of efficacy or contraindications to biologics [6]. Dalbavancin appears to be a promising treatment for late-stage HS with predominantly isolated Gram-positive bacterial strains [6].

### 3.7. Linezolid

Linezolid is considered the first member of oxazolidinones, a class of antibiotics that inhibits the formation of the initial 70S complex, blocking the synthesis of protein [55]. The spectrum of action comprised Gram-positive and atypical microorganisms, including *M. tuberculosis* [4,56].

Only a case report about linezolid use in HS was published. Specifically, it described the case of a 57-year-old woman on dialysis with stage III HS, treated with linezolid 1.2 g plus meropenem 1 g after failing several antibiotic therapies. She obtained clinical remission but experienced recurrence two weeks after discontinuation of intravenous therapy [4,57]. Currently, there is no definite evidence on the use of linezolid in the treatment of HS.

## 4. The Choice of Antibiotics Therapy in Clinical Practice

The role of bacterial infections and dysbiosis in the pathogenesis of HS has not been fully clarified, but they seem to contribute to an inflammatory vicious cycle underlying the pathophysiology of disease. In HS acute flare, polymorphous anaerobic and aerobic bacteria were frequently isolated in HS lesions [58].

Antibiotic therapy represents the mainstay of HS management, regardless of microbiologic screening. Currently, the real utility of swab and culture tests is debated and controversial. By definition, a negative culture may support a clinical diagnosis of HS [19].

In order to obtain disease remission, several conditions should be contemplated in the choice of antibiotic: patient characteristics (age, comorbidities, disease history, social class), disease features (severity, affected sites, first-line vs subsequent lines), and drug qualities (type of administration, previous treatments, cross-resistances, toxicity profile) [59] (Table 3).

Patients with severe disease may benefit from the combination of medical treatments (biologic or antibiotics) with surgery, which should be performed during disease remission. Appropriate antibiotic therapy could be performed to reduce inflammation, preparing the patient for surgical treatment [10].

### 4.1. The Treatment of HS with Antibiotics across Guidelines

According to guidelines, disease management is gradual and should consider the severity and the extension of HS (Figure 2).

Topical antibiotic and/or antiseptic therapy are recommended in patients with mild stage HS, localized disease, or in combination with systemic therapy. Topical clindamycin 1% is recommended across all guidelines as a first-line modality for HS treatment (mentioned in the North American, European S1, Canadian Association of Dermatology, Canadian Consensus, European HS Foundation and Brazilian guidelines) [10].

Resorcinol 15%, which has antimicrobic, anti-inflammatory, and keratolytic activities, have been reported as a valid alternative in short and long-term management of non-fistulous and fistulous HS lesions [25,26].

In severe and disseminated diseases, when the application of topical agents is difficult and impractical, systemic antibiotic therapy is suggested. Oral tetracyclines are recommended as first-line therapy in mild-to-moderate HS, especially when lesions are widespread or involve multiple anatomic sites. Regarding the duration of treatment, several recommendations are reported in different guidelines. The British, North American, HS ALLIANCE, and Brazilian guidelines suggest a treatment period of 12 weeks [19,60,61,62]. British guideline recommends discontinuing the treatment after 12 weeks to reduce the risk of antibiotic resistance [60]. Conversely, the European HS Foundation and European S1 guidelines recommend the use of oral tetracyclines for up to 4 months, while the Swiss guideline suggests a 3–6-month course [63,64]. North American guidelines also discuss the opportunity of maintenance therapy with oral tetracyclines.

The combination clindamycin–rifampicin is used as a second-line treatment for mild-to-moderate HS unresponsive to topical agents and oral tetracyclines. The efficacy of combination therapy is thought to be related to the mechanism of action of rifampicin, while clindamycin is added to limit rifampicin resistance [65]. Prolonged use of clindamycin and rifampicin beyond 10 weeks was found to be safe, but extended therapy is burdened by the risk of developing resistances to the first-line drug for the treatment of tuberculosis. Thus, as suggested by Brazilian guidelines, judicious is require [62,65,66].

Third-line therapies include combination therapies, as the triple association of metronidazole (500 mg TID), moxifloxacin (400 mg/day), and rifampicin (300 mg BID or 10 mg/kg day) for 6 weeks, indicated in patients with moderate–severe HS, according to the North American and European guidelines S1, and in patients with mild–moderate HS, as recommended by the HS ALLIANCE guidelines. Metronidazole should be discontinued after 6 weeks to avoid neurotoxicity [19,67,68].

As indicated in ALLIANCE and North American HS guidelines, intravenous ertapenem (1 g/day or 6 weeks) is an option for HS unresponsive to oral antibiotics, as rescue therapy or as a bridge therapy to surgery. Although its use is limited by intravenous administration, ertapenem resulted in the remission of Hurley stage I/II disease and significant improvement in quality of life in Hurley stage III patients [19,40,61].

Although the main guidelines (North America, South America, and Europe) strongly support the use of topical clindamycin, oral tetracyclines, and the combination of clindamycin and rifampicin as first-line treatment, significant discrepancies between second- and third-line treatment options emerged, relating to the absence of large-scale and high-quality studies. In addition, in all guidelines smoking cessation and weight reduction are encouraged as adjuvant therapy for HS. High BMI and smoking habits seem to predict a poor therapeutic response [10,69].

### 4.2. Limitations of Antibiotics in HS: Resistances and Toxicity

The treatment of HS is burdened by the occurrence of acquired antibiotic resistances. A prospective study on 69 patients and resistant strains to lincosamides and tetracyclines were isolated in 50% and 65% of patients with HS, respectively. Moreover, an analysis of 239 patients with HS from 2010 to 2015 showed that patients treated with topical clindamycin were more likely to have clindamycin-resistant *S. aureus* grow (63%) than patients treated without the antibiotic (17%) [70]. Bettoli et al., evaluating purulent material collected from 137 skin lesions of HS patients, reported the growth of 163 bacteria (55% Gram-positive and 44% Gram-negative). The most prevalent antibiotic resistances observed were to clindamycin (65.7%), rifampicin (69.3%), penicillin (70.0%), ciprofloxacin (74%), tetracycline (84.7%), and erythromycin (89.0%) [58].

Rifampicin represents the cornerstone of antibiotic combination therapy of tuberculosis. The use of rifampicin in patients with latent tuberculosis may enhance the occurrence of resistance. Thus, systematic tuberculosis screening is recommended in all patients with HS before starting treatment with rifampicin [71].

Concerning toxicity, gastrointestinal side effects (diarrhea, dyspepsia, nausea) are common in patients receiving tetracyclines (25–40%), ertapenem and clindamycin. Other treatment-related adverse events include skin manifestations (rash and itching, photosensitivity) and flu-like symptoms, related to tetracyclines; itching, flushing, and headaches, related to the infusion of dalbavancin; anemia related to linezolid, liver injury, and worsening renal function, associated to rifampicin; metronidazole-related neutropenia [59,65].

Notably, systemic clindamycin represents a major cause of pseudomembranous colitis. However, it is rarely described in HS patients for the association to rifampicin that seems to reduce the occurrence of colitis. The pathomechanism is largely unknown. The activity of cytochrome-P450’s induced by rifampicin seems to reduce the circulating concentrations of clindamycin [65]. Moreover, physicians should consider the potential drug–drug interactions. Additionally, a treatment regimen of rifampicin monotherapy (300 mg BID) for the first 7 days and the addition of clindamycin (300 mg BID) in the following days has been proposed as a strategy to reduce side-effects and improve patients’ compliance [72].

Tetracyclines should be avoided in pregnant women for the risk of tooth discoloration and interference with bone growth, although a recent systematic review does not support this association [10,73].

All these adverse events in association with the comorbidities, should be considered before the antibiotic prescription. We summarize the main toxicities related to systemic antibiotics (Table 3).

According to these two limitations, bacterial resistances and toxicity, maintenance with antibiotic therapy, especially clindamycin and rifampicin, should be avoided. However, it should be highlighted that the greatest risk of adverse effects from this combination occurs in the first few weeks of treatment. Conversely, tetracyclines (or macrolides) are potential candidates for maintenance regimens, particularly for their anti-inflammatory activity [28,29,30,65].

However, the choice of antibiotic is often based on personal experience or the popularity of a specific antibiotic, rather than on literature data or current guidelines. The empirical approach should be firmly discouraged, and novel therapeutical approaches should be evaluated in clinical trials [74].

## 5. Current Issues and Future Challenges

The most urgent issue related to antibiotic treatment in HS is represented by the increasing burden of antibiotic resistances and cross-resistances, which has been extensively documented. Among Gram-positive pathogens (typical findings of HS lesions), a global pandemic of resistant species, including *S. aureus* and *Enterococcus* spp., are now the greatest threats [45,75]. Resistance from *S. pneumonia* and *M. tuberculosis* (including isoniazid and rifampicin) is becoming epidemic. Additionally, Gram-negative pathogens are becoming resistant to several available antibiotic drug options [4]. However, in a retrospective study including 4.919 patients with HS and a history of antibiotic use, the majority of oral antibiotics for HS were administered in less than 90 days (12 weeks), which is consistent with guidelines for HS and other dermatological conditions [76].

A common mechanism of antibiotic resistance is represented by the production of biofilm [77]. It has been demonstrated that 89% of *S. epidermidis* isolated in patients with active HS lesions are strong biofilm producers in vivo [78]. While tetracycline and clindamycin are less active in biofilms, rifampicin represents the most effective antibiotic in biofilm eradication [79]. Currently, bactericidal lipoglycopeptide and glycopeptide antibiotics, such as daptomycin, teicoplanin and dalbavancin, have a significant role in therapy against Gram-positive bacterial infections and are usually recommended for the treatment of complicated skin, soft tissue, and bloodstream infections caused by *S. aureus* [53,80].

Other possible approaches active against biofilm include laser-therapy and surgery [19,81,82,83,84].

As stated, according to current guidelines, the choice of antibiotic should not be swab-guided because the clinical utility of this approach has not been demonstrated. The antibiogram may reveal non-standard antibiotics characterized by low minimum inhibitory concentration (MIC) and avoid useless toxicities in case of resistance. Moreover, the skin bacterial microflora is heterogeneous, and the identification of the most involved strain would be difficult. Therefore, the clinical utility of swab-guided antibiotic therapy in HS should be evaluated in terms of toxicity and activity in a randomized clinical trial [19].

The lack of a standard of care as a third-line treatment represents another unmet need. Several therapeutic options outside the international guidelines are available and a personalized approach is usually preferred. Therefore, there is an urgent need to collect prospective and robust data on this issue, assessing the efficacy of antibiotic treatments in clinical trials. No clinical trial on antibiotic therapy in HS is currently ongoing [85].

## 6. Conclusions

Biologics and small molecules have revolutionized the therapeutic armamentarium of several immune-mediated inflammatory cutaneous disease, such as psoriasis, atopic dermatitis, and HS [86,87,88,89,90]. To date, adalimumab remains the only European Medicines Agency (EMA)- and Food and Drug Administration (FDA)-approved biologic for the management of moderate-to-severe HS [91,92].

However, in the last decade, our understanding of the pathogenetic pathways that drive HS is rapidly emerging and the research of novel therapeutic targets have been intensified. Several cytokines have been identified in HS lesional skin, but the precise cytokine profile and the exact sequence of the inflammatory cascade are still to be elucidated. The use of biologics and small molecules in HS accelerates our understanding as they present the opportunity to highlight relevant cytokines patterns based on the clinical effectiveness of the biologic drugs. The results from clinical trials on biologics targeting TNF-α, IL-1, and IL-17 in HS are promising and although the number of effective biologic agents currently under investigation is encouraging, future treatment should be guided by personalized therapy, biomarkers, and pharmacogenomics. In addition, recent studies focusing on the characterization of the microbiome, proteome and transcriptome in HS are emerging. In this panorama, we believed that the combination of medical therapy with biologics and surgical therapy could have a key role in the management of the disease. However, in our opinion, antibiotics remains a valid therapeutic approach in HS particularly as supportive therapy in selected patients in order to control disease flare; as bridge therapy with surgical management; in association with biologic therapy; or in the case of contraindication to the biologic drug. The most important current issues are the prevention of antibiotic resistance and the definition of a standard antibiotic therapy beyond the second line. Therefore, the exploitation of novel strategies and combinations is warranted, especially in well-designed prospective observational studies or randomized control trials.

## Figures and Tables

**Figure 1 antibiotics-12-00978-f001:**
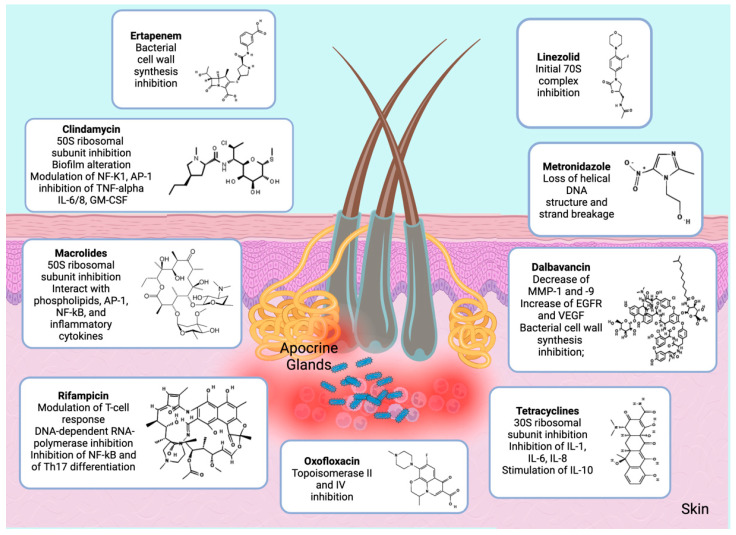
Mechanism of action of antibiotics used in HS. Keys: AP-1: Activator protein-1; EGFR: Epidermal growth factor receptor; GM-CSF: Granulocyte macrophage colony-stimulating factor; IL-1/6/8/10: Interleukin-1/6/8/10; MMP-1/9: Matrix metallopeptidase-1/9; NF-κb: Nuclear factor kappa-light-chain enhancer of activated B cells; Th17 cell: T helper 17 cell; TNF-α: Tumor necrosis factor-alpha; VEGF: Vascular-endothelial growth factor.

**Figure 2 antibiotics-12-00978-f002:**
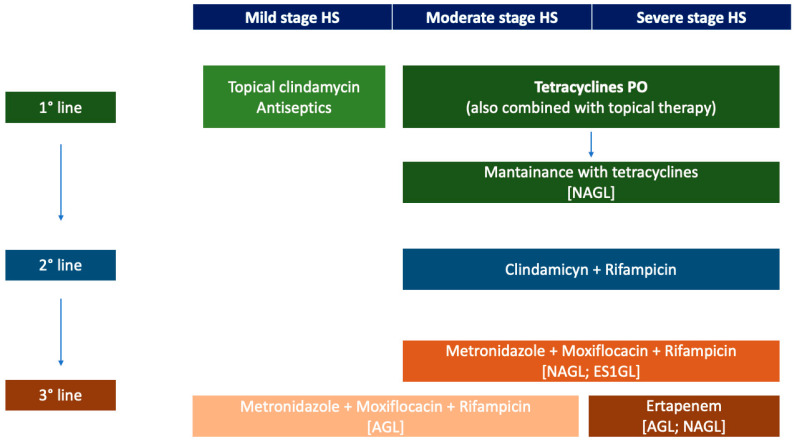
First, second, and third lines of antibiotic treatment in HS. Keys: AGL: ALLIANCE HS guidelines; ES1GL: European S1 guidelines; HS: hidradenitis suppurativa; NAGL: North American guidelines; PO: per os.

**Table 1 antibiotics-12-00978-t001:** Main tools commonly used in the assessment of HS severity and disease burden.

Clinical and Research Settings			
**International Hidradenitis Suppurativa Severity Score System (IHS4)**			
IHS4 (point) = - Number of nodules × 1 - Number of abscesses × 2 - Number of draining tunnels (fistulae/sinuses) × 4			
Mild HS: ≤3 points Moderate HS: 4–10 points Severe HS: ≥11 points			
Advantages		Limitation	
- It is a dynamic HS score. - It is simple to calculate. - Early identification of patients with moderate and severe disease, based on the presence of a draining tunnel (fistula/sinus) which is sufficient to classify an HS patient as at least moderate case.		- It is exclusively physician-based.	
**Hurley Staging System**			
Stage	I	II	III
Abscess	Single or multiple	Single or multiple, widely separated, recurrent	Diffuse or near-diffuse involvement
Sinus tract	–	+	Multiple interconnected
Cicatrization	–	+	+
Area			Entire area
Advantages		Limitation	
- It was originally designed for selection of the appropriate treatment modality in a specific body location (medical therapy for Hurley stage I, local surgery for Hurley stage II, and wide surgical excision for Hurley stage III). - It is reliable for rapid assessment of HS, and it is best for assessing Hurley stage III, meaning whether a patient should be candidate for surgery.		- It is static and it was not designed as a dynamic score for an accurate assessment of the extent of inflammation within each stage.	
**Hidradenitis Suppurativa Clinical Response—HiSCR**			
HiSCR is defined by the status of three types of criteria lesions, considering abscesses (fluctuant, with or without drainage, tender or painful), inflammatory nodules (tender, erythematous, pyogenic granuloma lesion), and draining fistulas (sinus tracts, with communications to skin surface, draining purulent fluid). The proposed definition of responders to treatment (HiSCR achievers) is: At least a 50% reduction in ANs; No increase in the number of abscesses; No increase in the number of draining fistulas from baseline.			
Advantages		Limitation	
- It is the most validated dynamic physical measure for assessing treatment response in RCTs.		- It has lower utility in the clinical setting.	
**Hidradenitis Suppurativa Physician’s Global Assessment (HS-PGA)**			
HS-PGA			
Clear (score = 0)		0 abscesses, 0 draining fistulas, 0 inflammatory nodules, and 0 non-inflammatory nodules	
Minimal (score = 1)		0 abscesses, 0 draining fistulas, 0 inflammatory nodules, and presence of non-inflammatory nodules	
Mild (score = 2)		0 abscesses, 0 draining fistulas, and 1–4 inflammatory nodules; or 1 abscess or draining fistula and 0 inflammatory nodules	
Moderate (score = 3)		0 abscesses, 0 draining fistulas, and 1 ≥ 5 inflammatory nodules; or 1 abscess or draining fistula and ≥1 inflammatory nodule; or 2–5 abscesses or draining fistulas and <10 inflammatory nodules	
Severe (score = 4)		2–5 abscesses or draining fistulas, and ≥10 inflammatory nodules;	
Advantages		Limitation	
- It is a dynamic HS score. - It is simple to calculate.		- It is lower utility in the clinical setting compared to RCTs.	
**Patient-Reported Outcomes**			
**Dermatology Life Quality Index (DLQI Score)**			
DLQI Total score			
0–1		No effect on patient’s life	
2–5		Small effect on patient’s life	
6–10		Moderate effect on patient’s life	
11–20		Very large effect on patient’s life	
21–30		Extremely large effect on patient’s life	
Advantages		Limitation	
- It is a valuable adjunct and straightforward to perform in clinical settings.		- It is a skin-specific questionnaire that poorly considers the pain/discomfort dimension, which is dominant in HS.	
**Pain Visual Analog Scale (Pain VAS) and Numeric Rating Scale (NRAS) for Pain**			
The pain VAS is a continuous scale comprised of a horizontal (HVAS) or vertical (VVAS) line, usually 10 cm in length, anchored by 2 verbal descriptors, one for each symptom extreme. The NRAS for pain is a single 11-point numeric scale, which can be administered verbally or graphically. **Response options/scale.** For pain intensity, both scales are most commonly anchored by “no pain” (score of 0) and “pain as bad as it could be” or “worst imaginable pain” (score of 10).			
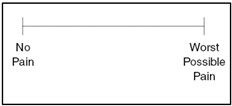		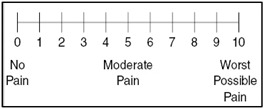	
Advantages		Limitation	
- VAS and NRAS are valuable adjuncts and straightforward to perform in clinical settings. - VAS and NRAS can be used to measure worst, least, or average pain over the last 24 h, or during the last week.		- While the VAS can be quick to use, it is not as practical as the Numerical Rating Scale (NRS) as it requires clear vision, dexterity, paper, and pen.	

**Table 3 antibiotics-12-00978-t003:** Toxicities related to the use of systemic antibiotics (based on the prescribing information).

Types of Toxicity	Antibiotic Drugs
Liver toxicity	Clindamycin (common); Ertapenem (uncommon); Macrolides (uncommon); Rifampicin (unknown); Tetracyclines (unknown); Metronidazole (uncommon); Linezolid (common).
Renal toxicity	Clindamycin (common); Ertapenem (uncommon); Macrolides (uncommon); Rifampicin (unknown); Oxofloxacin (uncommon); Metronidazole (uncommon); Linezolid (common).
Gastrointestinal toxicity	Clindamycin (common); Ertapenem (common); Macrolides (very common); Rifampicin (common); Oxofloxacin (common); Tetracyclines (common); Dalbavancin (common); Metronidazole (unknown); Linezolid (common).
Nervous system toxicity	Clindamycin (uncommon); Ertapenem (common); Macrolides (common); Oxofloxacin (uncommon); Rifampicin (common); Tetracyclines (common); Dalbavancin (common); Dalbavancin (uncommon); Linezolid (common).
Hemolymphopoietic toxicity	Clindamycin (unknown); Ertapenem (uncommon); Macrolides (uncommon); Rifampicin (unknown); Oxofloxacin (uncommon); Tetracyclines (unknown); Dalbavancin (uncommon); Metronidazole (uncommon); Linezolid (common).
Vascular toxicity	Clindamycin (common); Ertapenem (common); Rifampicin (unknown); Oxofloxacin (rare); Dalbavancin (uncommon); Linezolid (common).
Cardiac toxicity	Clindamycin (uncommon); Ertapenem (uncommon); Macrolides (uncommon); Oxofloxacin (rare).
Cutaneous and subcutaneous toxicity	Clindamycin (common); Ertapenem (common); Macrolides (uncommon); Rifampicin (unknown); Oxofloxacin (common); tetracyclines (unknown); Dalbavancin (uncommon); Metronidazole (uncommon); Linezolid (common).
Infections	Clindamycin (unknown); Ertapenem (uncommon); Macrolides (uncommon); Rifampicin (unknown); Oxofloxacin (uncommon); Dalbavancin (uncommon); Metronidazole (common); Linezolid (common).
Ototoxicity	Macrolides (uncommon); Linezolid (uncommon).
Musculoskeletal toxicity	Macrolides (uncommon); Rifampicin (unknown); Oxofloxacin (uncommon); Metronidazole (uncommon).

## Data Availability

Not applicable.

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
