# Peer review of "Systemic Antibiotic Therapy in Hidradenitis Suppurativa: A Review on Treatment Landscape and Current Issues"

_antibiotics, 2023, doi:10.3390/antibiotics12060978_

Round 1

Reviewer 1 Report

The authors should consider the followings:

In Table 1, in the column of study design endpoint, the authors should label the primary endpoint(s) across the studies, where available.

The authors can analyze the risk of bias in this review topic, to supplement the current section of limitions.

The authors may make a brief comparison table of the multiple tools, in the assessment of HS severity and disease burden, and present both the advantages and limitations of them.

In Table 1, in the column of outcome, the authors may include the results of OR, RR, HR etc to overview the effect size, if available.

The authors may provide a figure to list the chemical structures of reviewed antibiotics in this review.

The authors can briefly describe the search strategies and database used, and range years of publications in this review.

The authors may tabulate the types of toxicity (liver, renal, oto-, etc), under the section of antibiotics induced toxicity, and whether the toxicity can be limited by topical application vs systemic.

The authors shall explain on any major updates regarding the current review, to that of Marasca et al 2020. The Pharmacology of 470 Antibiotic Therapy in Hidradenitis Suppurativa. Expert Rev Clin Pharmacol, 13, 521–530.

Author Response

REVIEWER 1

  1. In Table 1, in the column of study design endpoint, the authors should label the primary endpoint(s) across the studies, where available. We added the primary endpoint(s) of the study in the Table 1, as required.
  2. The authors can analyse the risk of bias in this review topic, to supplement the current section of limitations. The risk of quantitative bias (ROB assessment) is not applicable to narrative reviews. We used SANRA scale on preparing our narrative reviews to improving the quality of submitted manuscript.

(Baethge C, Goldbeck-Wood S, Mertens S. SANRA-a scale for the quality assessment of narrative review articles. Res Integr Peer Rev. 2019 Mar 26;4:5.) We added in conclusion the limitation of this type of review.

  1. The authors may make a brief comparison table of the multiple tools, in the assessment of HS severity and disease burden, and present both the advantages and limitations of them. We added a table summarizing the several tools that were used in the assessment of HS (disease severity and quality of life). (Table 1)
  2. In Table 1, in the column of outcome, the authors may include the results of OR, RR, HR etc to overview the effect size, if available. We added OR, RR, HR (if available) in the Table 2, as required.
  3. The authors may provide a figure to list the chemical structures of reviewed antibiotics in this review. We have included the chemical structures of antibiotics in Figure 1.
  4. The authors can briefly describe the search strategies and database used, and range years of publications in this review. We added the search strategies and database used, as required.

  1. Materials and methods

Bibliographic searches for qualitative review were conducted in PubMed up to February 20, 2023, with no date limits, by using the terms: (hidradenitis suppurativa OR acne inversa OR Verneuil’s disease) AND (antibiotic), (lincosamides), (clindamycin), (rifampicin), (tetracycline), (lymecycline), (doxycycline), (beta-lactam antibiotics), (penicillins), (cephalosporins), (carbapenems), (monobactams), (ertapenem), (macrolides), (azithromycin), (clarithromycin), (erythromycin), (metronidazole), (glycopeptide), (dalbavancin), (linezolid). One author (EDS) initially screened all titles and abstracts and excluded articles that were clearly ineligible. Reports and cases were excluded if clinical details were lacking. When eligibility was in doubt, other two authors (EM) and (MC) were involved. Articles were limited to those in English language. Full texts of the included articles were reviewed, and reference lists were manually searched and were checked for additional sources.

  1. The authors may tabulate the types of toxicity (liver, renal, oto-, etc), under the section of antibiotics induced toxicity, and whether the toxicity can be limited by topical application vs systemic. We added a table that includes the type of toxicity related antibiotics. (Table 3). Although topical administration is characterized by lower toxicity compared to systemic administration, no data comparing topical versus systemic administration are available of the same antibiotic.

  1. The authors shall explain on any major updates regarding the current review, to that of Marasca et al 2020. The Pharmacology of Antibiotic Therapy in Hidradenitis Suppurativa. Expert Rev Clin Pharmacol, 13, 521–530. This review was structured according to antibiotics mechanism of action. Updated data were reported (e.g. data on the use of dalbavancin and metronidazole). In addition, we discussed the choice of antibiotic in clinical practice and the current guidelines. Lastly, we further discussed the issue of antibiotic resistance.

Reviewer 2 Report

1-    Please indicate the type of study in the title. Is this a review article? Retrospective cohort study or what?

2-    Make the abstract structured based on the journal style 

3-    The introduction is many small paragraphs that may be not appealing in reading. I can see so many paragraphs that can be combined together. Please revise critically 

4-    Table 1. You are summarizing the studies evaluating different treatments. You should make an additional column that should be the first column “Author” or “study” for example James et al. 

5-    I like the tables and figures

6-    Similar to my suggestion in the introduction. Try to avoid many small paragraphs and combine them whenever possible 

Author Response

REVIEWER 2

  1. Please indicate the type of study in the title. Is this a review article? Retrospective cohort study or what? We modified the title as follow: Systemic antibiotic therapy in hidradenitis suppurativa: a review on treatment landscape and current issues.

  1. Make the abstract structured based on the journal style.

Abstract:

Hidradenitis suppurativa (HS) is a chronic, recurrent, inflammatory skin disease characterized by painful, deep-seated, nodules, abscesses and sinus tracts in sensitive areas of the body, including axillary, inguinal, and anogenital regions. Antibiotics represent the first-line pharmacological treatment of HS, for their anti-inflammatory properties and antimicrobial effects. This narrative review summarizes the most significant current issues on the role of systemic antibiotics in the management of HS, critically analyzing the main limits of their use (antibiotic resistance and toxicity). Although in the last decades several cytokines have been implicated in the pathomechanism of HS and the research on the use of novel biologic agents in HS has been intensified, antibiotics remain a valid therapeutic approach. Future challenges about antibiotic therapy in HS comprise their use in association to biologics in the management of acute flare or as a bridge therapy to surgery.

  1. The introduction is many small paragraphs that may be not appealing in reading. I can see so many paragraphs that can be combined together. Please revise critically. We have revised the introduction, as suggested.

  1. Table 1. You are summarizing the studies evaluating different treatments. You should make an additional column that should be the first column “Author” or “study” for example James et al. We have added a column reporting the author of the study, as suggested.

  1. I like the tables and figures. Thanks for the comment.

  1. Similar to my suggestion in the introduction. Try to avoid many small paragraphs and combine them whenever possible. We have revised all the manuscript, delating the small paragraphs, as required.

Reviewer 3 Report

1. The strong opinion towards the treatment of HS was not mentioned.

2. Conclusion of the study needs to be clear on the authors view.

3. Additionally, what is the significance of bactericidal lipoglycopeptide and glycopeptide antibiotics in treating Gram-positive bacterial infections? Please explain more about it. and it can be a reference for using it

Author Response

REVIEWER 3

  1. The strong opinion towards the treatment of HS was not mentioned. We have mentioned in the conclusion of the manuscript.
  2. Conclusion of the study needs to be clear on the author’s view. We revised the conclusion of the manuscript, clarifying author’s point of view.

  1. Additionally, what is the significance of bactericidal lipoglycopeptide and glycopeptide antibiotics in treating Gram-positive bacterial infections? Please explain more about it. and it can be a reference for using it. We have elucidated the significance of bactericidal lipoglycopeptide and glycopeptide antibiotics in treating Gram-positive bacterial infections and we added the appropriate bibliographic references. (Van Bambeke F, Van Laethem Y, Courvalin P, Tulkens PM. Glycopeptide antibiotics: from conventional molecules to new derivatives. Drugs. 2004;64(9):913-36.)

Round 2

Reviewer 2 Report

None